# Diversity of Microbial Mats in the Makgadikgadi Salt Pans, Botswana

**DOI:** 10.3390/microorganisms12010147

**Published:** 2024-01-11

**Authors:** Sevasti Filippidou, Alex Price, Charlotte Spencer-Jones, Anthony Scales, Michael C. Macey, Fulvio Franchi, Lesedi Lebogang, Barbara Cavalazzi, Susanne P. Schwenzer, Karen Olsson-Francis

**Affiliations:** 1AstrobiologyOU, Faculty of Science, Technology, Engineering and Mathematics, The Open University, Milton Keynes MK7 6AA, UK; s.filippidou@imperial.ac.uk (S.F.); a.price@microbiologysociety.org (A.P.); charlotte.l.spencer-jones@durham.ac.uk (C.S.-J.); anthony.scales@open.ac.uk (A.S.); michael.macey@open.ac.uk (M.C.M.); susanne.schwenzer@open.ac.uk (S.P.S.); 2School of Life Sciences, Imperial College London, London SW7 2AZ, UK; 3Department of Geography, Durham University, Durham DH1 3LE, UK; 4Earth and Environmental Science Department, Botswana International University of Science and Technology, Palapye 10071, Botswana; franchif@biust.ac.bw; 5School of Geosciences, University of the Witwatersrand, Johannesburg 2001, South Africa; 6Department of Biological Sciences and Biotechnology, Botswana International University of Science and Technology, Palapye 10071, Botswana; lebogangl@biust.ac.bw; 7Department of Biological, Geological, and Environmental Sciences, University of Bologna, 40126 Bologna, Italy; barbara.cavalazzi@unibo.it; 8Department of Geology, University of Johannesburg, Johannesburg 2006, South Africa

**Keywords:** salt pans, metagenomics, microbial mats, Makgadikgadi Basin

## Abstract

The Makgadikgadi Salt Pans are the remnants of a mega paleo-lake system in the central Kalahari, Botswana. Today, the Makgadikgadi Basin is an arid to semi-arid area receiving water of meteoric origin during the short, wet season. Large microbial mats, which support primary production, are formed due to desiccation during the dry season. This study aimed to characterise the microbial diversity of the microbial mats and the underlying sediment. The focus was the Ntwetwe Pan, located west of the Makgadikgadi Basin. Metagenomic analyses demonstrated that the mats consisted of a high relative abundance of Cyanobacteriota (synonym Cyanobacteria) (20.50–41.47%), Pseudomonadota (synonym Proteobacteria) (15.71 to 32.18%), and Actinomycetota (synonym Actinobacteria) (8.53–32.56%). In the underlying sediments, Pseudomonadota, Actinomycetota, and Euryarchaeota represented over 70% of the community. Localised fluctuations in water content and pH did not significantly affect the microbial diversity of the sediment or the mats.

## 1. Introduction

Salt pans are ubiquitous in arid and semi-arid environments where evaporation exceeds precipitation [1,2]. They are formed in closed basins and, in most cases, are the remnants of ancient paleo-lakes. The Makgadikgadi Salt Pans in the Central Kalahari Desert are the remnant of a mega paleo-lake formed in the Early Pleistocene from the southwest propagation of the East Africa Rift [3,4]. The drying out of the mega paleo-lake left a system of salt pans within the Makgadikgadi Basin, with the largest being the Sua (in the east), Ntwetwe (in the west), and Nxai (in the northwest) pans [5,6]. The Makgadikgadi Pans receive a mean annual rainfall of 300 mm, with seasonal rivers forming during the wet season (December to March) [7,8]. The area is characterised by strong seasonal and daily temperature fluctuations, with a maximum temperature of 48 °C in the summer and a minimum of 3 °C in the winter [9]. The long dry season (April to October) results in evaporation and deposition of evaporite minerals, mainly containing NaCl [7,9]. The immediate sub-surface of the Ntwetwe Pan is composed of silcretes and calcretes, characteristic of arid environment diagenetic processes [10].

Salt pans are generally nutrient-rich and highly productive when in flood due to the high temperatures and intense light, which drives phototrophic metabolism [11]. Using microbial culturing techniques, bacteria, such as Cyanobacteriota and Bacilliota (synonym Firmicutes), archaea [12], and microscopic eukarya, including yeasts and fungi [13], have been isolated from the Makgadikgadi Salt Pans. To our knowledge, there have been no published diversity analyses of the Makgadikgadi Salt Pans; however, in other salt pan environments, studies have shown that microbial diversity decreases with salinity, which is associated with an increase in the proportion of archaea (e.g., [14,15]).

Upon desiccation during the dry season, microbial mats form on the surface. Photosynthetic mats usually have an upper layer dominated by Cyanobacteriota, which play a key role in primary production and nitrogen fixation. As a by-product, they produce polymeric substances (Extracellular Polymeric Substances (EPS)), which bind sediment and minerals, providing physical protection and resistance to desiccation [16,17]. The diversity of the microbial mats is influenced by environmental stressors, such as salinity, seasonal desiccation, and high solar irradiance [18]. Thomas et al. [19] reported that salt-cyanobacteriota crusts north of the Ntwetwe Pan play a significant role in soil CO_2_ efflux.

In this study, we used a metagenomic approach to provide preliminary insight into the microbial diversity within the Makgadikgadi Salt Pans. The focus of the study was the microbial mats and sediment of the Ntwetwe Pan. To our knowledge, this is the first taxonomic analysis of the microbial mats from the Ntwetwe Pan.

## 2. Materials and Methods

### 2.1. Site Description and Sampling

Sediment and mat materials were collected from a region of the Ntwetwe Pan that was dominated by surface mats (20°35.506′ S, 25°31.331′ E) (Figure 1) in January 2020. The mats were thick crusts (maximum 5 mm thick) (Figure 2A–D) and could be easily sampled and separated from the underlying pavement (please see [3], for description). The mats (*n* = 9) and underlying sediment (*n* = 9) (1–10 cm in depth) were sampled using a trowel that was sterilised between sampling with Azo^TM^ wipes. A surface sediment sample (with no microbial mat) dominated by halite crystals was also collected (it was located approximately 1 m from the microbial mats). Upon collection, the samples were wrapped in pre-combusted (450 °C for 6 h) aluminium foil and stored at 4 °C before being shipped to the Open University, UK (the samples were stored at 4 °C for 7 d). On arrival at the laboratory, the sample materials were sub-sampled and stored for 14 d at −20 °C for DNA extraction (approximately 3 g of sediment and 1 g of microbial mat) and 4 °C (approximately 10 g of sediment and 1 g of microbial mat) for further sedimentological characterisation.

### 2.2. Geochemical Characterisation

Temperature (uncertainty of 0.5 °C) was measured in-situ using a Mettler Toledo FiveGo probe. The pH values of the sediment were verified using an Orion 3-Star Thermo Scientific benchtop pH meter (uncertainty of 0.01 pH units) at room temperature. For this, 5 g of sediment was suspended in Milli-Q water in a 1:2.5 ratio and shaken in a rotary shaker at approximately 120 rpm. After 2 h, the samples were removed from the shaker and analysed.

In addition, the bulk density and water content of the underlying sediment sample were calculated. For this, the wet sediment’s total surface area (mm) and total mass (g) were measured. The sediment sample was freeze-dried for 48 h, and then the dried weight and surface area were re-measured. The sediment-water content on a dry mass basis (g water/g of dry sediment) [20] and bulk density were calculated as detailed in Equations (1) and (2).
(1)$$Sediment\;water\;content\left(g/g\right)=\frac{mass\;of\;wet\;sediment-mass\;of\;dry\;sediment}{mass\;of\;dry\;sediment}$$
(2)$$Bulk\;density=\frac{mass\;of\;dry\;sediment}{total\;volume\;of\;sediment}$$

Total Organic Carbon (TOC) and Total Nitrogen (TN) were measured with an Elementar–VarioMax (using services offered by the British Geological Survey (BGS), Wallingford (UK)). Before analysis, the samples were acidified with hydrochloric acid to remove any carbonate.

### 2.3. Micromorphology

Images of the microbial mats were prepared using Transmission Electron Microscopy (TEM) and Field Emission Gun Scanning Electron Microscopy (FEG-SEM) with an X-ray Energy Dispersive Spectroscopy detector (EDS, Oxford Instruments, Abingdon, UK). For TEM, the sample preparation protocol was adapted from a previously described protocol [21]. The resulting block was sectioned (100 nm thick) using a diamond saw and collected on a copper grid coated with a pioloform support film, which was stained to enhance contrast in 3% uranyl acetate (30 min) and Reynolds lead citrate (10 min). The sections were imaged with a JEM 1400 (Jeol, Tokyo, Japan) at an acceleration voltage of 80 kV [22].

For FEG-SEM, the samples were fixed in 2.5% glutaraldehyde with 0.1 M sodium cacodylate buffer (pH 7.4), dyed with osmium, and dehydrated using the critical point drying method [23]. The samples were mounted on an aluminium stub using a carbon adhesive disc, coated with 20 nm carbon in a Safematic carbon coater (Labtech, Heathfield, UK). All Secondary Electron (SE) images were acquired at an accelerating voltage of 20 kV on a FEG-SEM (Zeiss Supra 55VP, Oberkochen, Germany). The EDS data were analysed using Oxford Instruments Aztec software (version 6.0). The elemental abundances deriving from the spectra were normalised to 100 wt %. The replicates were averaged for each particle studied, with at least five average particles per sample.

### 2.4. Mineralogical Composition

X-ray Diffraction (XRD) analysis was carried out to determine the dominant mineral phases in the sediment samples. Approximately 0.5 g of sample was crushed into a fine powder using a mortar and pestle. Samples were analysed using a Siemens D5000 instrument (New York, NY, USA) with a 0.04 degrees per step, 1 s per step from 5 to 100 (2-theta degrees). The radiation source was a Cu tube producing Cu k-alpha radiation at a wavelength of 1.5406 Å. The data generated were analysed using the QualX software (version 2.0) in combination with the bulk chemistry of the sample to identify crystalline phases [24].

### 2.5. Metagenomic Analysis

DNA was extracted from 0.5 g of sample material using the FastDNA^®^ SPIN Kit for Soil (MP Biomedicals, Solon, OH, USA), per the manufacturer’s instructions. The DNA was eluted using 100 μL of elution buffer. Purified DNA extracts were quantified with a Qubit^®^ 2.0 Fluorometer (Life Technologies Corporation, Carlsbad, CA, USA) using the Quant-iT dsDNA BR assay kit and following the manufacturer’s instructions. The Nextera DNA library preparation kit (Illumina, San Diego, CA, USA) was used for library preparation. Metagenome sequencing was performed with the MiSeq2500 platform (Illumina, San Diego, CA, USA) using the service provided by IGA Technology (Udine, Italy).

Sequences were uploaded to MG-RAST for sequencing analyses [25]. Low-quality sequencing was removed using DyanicTrim. The lowest phred score counted as a high-quality base was 15, and sequences were trimmed to contain at most five bases below the above-specified quality. The sequencing error of the metagenomic shotgun data was estimated using DRISEE [26]. The redundancy of the reads was assessed using the Nonpareil tool from the Enveomics suite [27]. This was carried out to calculate the average coverage and predict the number of sequences required to achieve complete coverage. MEGAHIT (version 1.2.9) performed de novo assembly with default parameters to obtain contigs (min 3000 bp) [28]. Potential rRNA gene sequences were identified by BLAST similarity searches against the M5rna database, which integrates SILVA [29], Greengenes [30], and the Ribosomal Database Project (RDP) [31]. The sequencing data was deposited to NCBI Sequence Read Archive (SRA) under the accession numbers SRR27256811-SRR27256830.

### 2.6. Statistical Analysis

All statistical analysis of metadata and metagenomic data was performed on R studio, version 1.4.1717, using Vegan [32], BiodiversityR [33], and Grid [34]. Data were visualised using ggplot2 [35], except the diversity bar charts that were visualised in MS Excel. Shannon, Simpson and Sørensen indices were calculated using Vegan to measure α-diversity. Correlations were calculated using Pearson’s method. The differences in community composition and environmental conditions were visualised using Principal Component Analysis (PCA). Variable, fixed, and NMDS distances were calculated using the Bray–Curtis method. *P*-values were calculated using Welch’s *t*-test, and only correlations with *p*-values < 0.01 were considered.

## 3. Results

### 3.1. Geochemistry

As shown in Figure 2A–D, the mats varied in colour due to their water content. This was reflected in the underlying sediment; for example, the mean water content was 0.251% for USA compared to 0.157% in USC, as shown in Figure 3. XRD analysis showed that the sediment samples were dominated by quartz (ranging between 53.50–100%) and calcite (0.00–40.40%) and confirmed the results presented in Franchi et al., 2022. The TOC and TN ranged between 0.45–1.50% and 0.02–0.13%, respectively (Table 1). In general, the TOC was significantly higher in the mats than in the underlying sediment, with no significant difference with the TN (Table 1). There was no significant difference between the pH of the sediment and the mats, with values ranging between pH 9.48 and 10.40.

### 3.2. Microbial Mat Structure

The microbial mats were tightly compacted and composed of filamentous and coccid-shaped cells located on a silica and calcite-dominated mineral substrate (Figure 4A). TEM analyses suggested that the microbial cells were located within a carbon-rich matrix produced by EPS (Figure 4B), with cyanobacterial cells surrounded by a mucilaginous sheath, as shown by the arrow in Figure 4C.

### 3.3. Microbial Community Composition

In total, 19 metagenomes were generated, obtaining an average of 11,980,303 reads per metagenome (average read length of 149 ± 8 bp) with the lowest phred score of 15. The overall taxonomic analysis of the metagenomic reads demonstrated a total of 67 Phyla, with a cut-off of 70% identity. Of these Phyla, five belonged to Archaea (Crenarchaeota, Euryarchaeota, Korarchaeota, Nanoarchaeota, and Thaumarchaeota), 26 to Bacteria, and 36 to Eukarya (seven of which were Fungi: Ascomycota, Basidiomycota, Blastocladiomycota, Chytridiomycota, Glomeromycota, Microsporidia, and Neocallimastigomycota).

As demonstrated in Figure 5, the most relevant abundant Phylum in the underlying sediments was Actinomycetota (mean 32.79%, ranging from 26.76 to 39.15%). In the mats, it was Cyanobacteriota (mean 31.32% ranging from 20.51 to 41.47%), followed by Pseudomonadota (mean 22.77% ranging from 15.71 to 32.18%). At the genus level, the most abundant genera in the mats were *Microcoleus* (Cyanobacteriota), *Rubrobacter (Actinomycetota), Truepera* (Deinococcota), *Streptomyces* (Actinomycetota). *Cyanothece* (Cyanobacteriota), *Nocardiopsis* (Actinomycetota), *Halalkalicoccus* and *Haloterrigena* (Euryarchaeota), *Conexibacter* (Actinomycetota), and Nostoc (Cyanobacteriota). This is compared to the underlying sediments, where the most abundant genera were *Rubrobacter*, *Truepera*, *Streptomyces*, *Halalkalicoccus*, *Conexibacter*, *Haloterrigena*, *Gemmatimonas* (Gemmatimonadota), *Mycobacterium* (Actinomycetota), *Natrialba* (Euryarchaeota), and *Frankia* (Actinomycetota). In the surface sample with no microbial mat, the most abundant genera were the halophilic Archaea *Natrialba*, *Haloterrigena*, *Natronomonas*, *Halalkalicoccus*, *Halorubrum*, *Haladaptatus*, *Haloarcula*, *Halorhabdus*, *Halogeometricum*, *Haloferax*, and *Halomicrobium*.

The abundance of archaea in the underlying sediment was significantly higher than in the mats. For example, they represented between 1.85 and 4.81% of the mats compared to between 8.28 and 19.33% of the underlying sediment communities. However, in the sample with no surface mat, Archaea represented 65.4% of the total microbial community, with Euryarchaeota the most prevalent Phylum, representing between 60.3 and 65.2% of the Archaea. Fungi represented only a small fraction of the total microbial community, <1% of the mats and <3% of the sediment communities. Viral sequences represented <0.07% of the total sequences and could not be further classified.

Diversity indices, including the Shannon Simpson and Sørensen, indicated that the mats were less diverse than the underlying sediments (Figure 6A–C, respectively). Environmental factors, including water content, pH, % of C, % of N, and C/N ratio, did not appear to significantly impact microbial diversity, with no correlation observed between diversity indices and the environmental factors shown in Table 1. Sorenson’s coefficient showed no significant difference between the mats and the underlying sediment (Figure 6D). PCoA using Bray–Curtis dissimilarities showed that the underlying sediments were more similar to each other than their corresponding mats.

## 4. Discussion

In this study, we investigated the microbial diversity of the Northern Ntwetwe Pan in the Makgadikgadi Salt Pans, Central Kalahari, Botswana. This area harbours extensive microbial mats interspersed with rock debris (Figure 2). The mats were not uniformly distributed throughout the studied area but instead form small regions of approximately 20 cm in diameter (Figure 2); however, there are examples of much larger biofilms reported in the other areas of the Makgadikgadi Salt Pan (e.g., [19]). It is important to highlight that samples were collected during the wet season, and the area received rain overnight; hence, the water content was arbitrary. However, the rainfall allowed discernment of areas that dry very quickly (light-toned areas in this study) from areas which stay moist for longer (darker-toned areas). This pattern was assumed to be stable due to its relationship with small-scale topography and sediment texture. TOC and TN did not differ among the sites, nor did their mineral composition, based on the XRD data, confirming previous studies in similar environments [36,37,38]. The cells within the microbial mats were embedded within an organic matrix, which was attributed to the production of EPS. This is common in microbial mats located in hypersaline environments; the EPS matrix can shield the cells from environmental stressors, such as desiccation, UV radiation, heavy metals, and antimicrobial compounds (as discussed in [39]).

The taxonomic analysis of the microbial communities revealed a ubiquitous distribution of bacterial Phyla: Actinomycetota, Pseudomonadota, Cyanobacteriota, and Bacilliota. These Phyla are known to be cosmopolitan and represent a significant component in soil and sediment microbial communities, especially desert soils [40]. Chloriflexota, Bacteriodota and Deinococcota were identified in lower abundance in all samples (below 6%). According to the statistical analysis, these Phyla (along with Cyanobacteriota that were predominantly only found in the mats and of low abundance in underlying sediments) were responsible for the major differences between these samples. Ordination analysis showed that these taxa impacted the correlations between species richness, α-diversity, and environmental parameters more than the most abundant taxa, Actinomycetota and Pseudomonadota.

At the Phylum levels, the community profile of the microbial mats showed a large overlap with profiles observed in microbial mats from other hypersaline lakes, for example, La Brava and Tebenquiche in the Salar de Atacama (Chile) [41], a desiccating pond in the Cuatro Cienegas Basin (Mexico) [42], saline springs in the Namib Desert (Namibia) [43], and Shark Bay (Australia) [44]; however, for the Namibian, Chilean and Australian environments, these mats were dominated by the same Phyla, but with a greater abundance of Bacteroidetota, representing a greater abundance of the genus *Salinibacter* as opposed to the halophilic members of the Euryarchaeota observed in this study. Microbial mats in Lake Medoza (Argentina) and Highborne Cay (Bahamas) also contained members of the Bacilliota and Pseudomonadota but were dominated by Euryarchoaeta and Cyanobacteriota, respectively, reflecting the impact that light and geochemical regimes can have on driving the community composition [45,46].

The α-diversity metrics showed that community diversity did not significantly change as the water content decreased within the microbial mats, contradicting previously observed data [47]. This is potentially because water was transient at this site, so the content differs throughout the year. This was also reflected in the underlying sediment. The Sørensen index demonstrated that the microbial mat communities were highly similar to their underlying sediment, and this similarity increased with desiccation (from 0.93 to 0.99). This could seem paradoxical since lower water content resulted in unsaturated environments. However, the filamentous nature of the mats could enhance movement and communication, as previously shown to occur for unsaturated soils [48,49].

The abundance of Archaea was greater (over 65.21%) in areas with no microbial mats. This is consistent with observations in other salt pan environments, for example, where Euryarchaeota, particularly Halobacteriales, have been shown to increase with salinity in saline soils [50]. Furthermore, a previous study of the West Kalahari (Witpan and Omongwa Pans) showed that Euryarchaeota represented over 65% of the microbial community, with Gemmatimonadota and spore-forming Bacilliota representing the most abundant bacterial phyla [51].

## 5. Conclusions

In conclusion, preliminary metagenomic analyses gave insight into the microbial diversity of the Makgadikgadi Salt Pans. Seasonal variation in evaporation and desiccation resulted in the formation of microbial mats, which protect the microbial community from environmental stresses and influences the diversity of the surface community. The microbial mats were dominated by the Cyanobacteriota and Pseudomonadota, whereas Euryarchaeota, dominated the surface sediment, which did not possess surface mats. Future work will focus on understanding the functional diversity and ecological roles of the microorganisms within the Makgadikgadi Salt Pans.

## Figures and Tables

**Figure 1 microorganisms-12-00147-f001:**
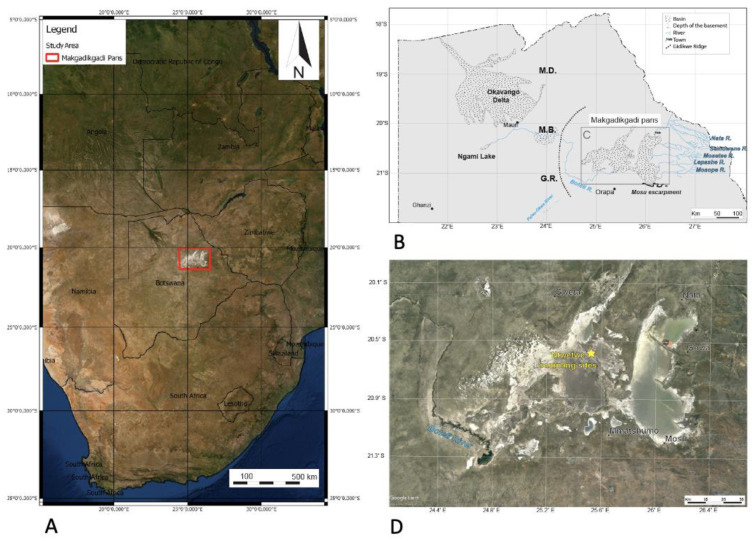
(**A**) Location of the Makgadikgadi salt pans in southern Africa. The red box outlines the area shown in (**C**) (modified from [6]). (**B**) Schematic view of the main basins in northern Botswana, showing the main affluent to the Makgadikgadi Pans and relevant geomorphic features. Ma—Makalamabedi Basin; G.R.—Gidikwe ridge; M.D.—Mababe Depression. (**D**) Satellite image of the Makgadikgadi Pans showing the sampling site (yellow) in the northern Ntwetwe Pan. Source Google Earth, Image Landsat/Copernicus, US Department of State Geographer, 2021.

**Figure 2 microorganisms-12-00147-f002:**
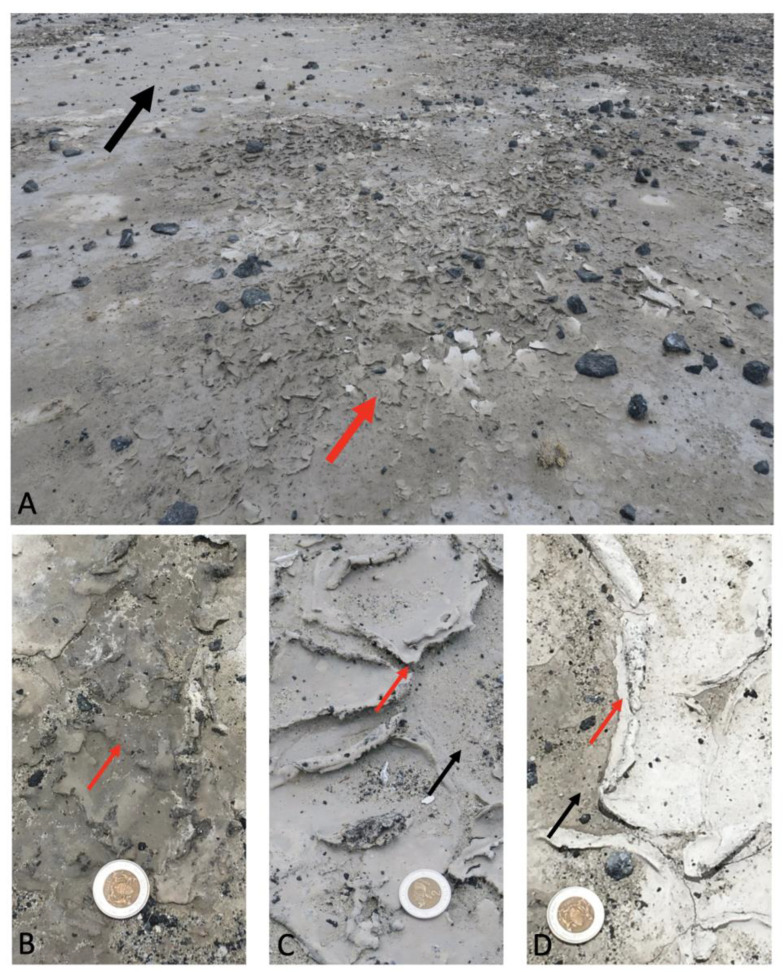
(**A**) The sampling site located in the Ntwetwe Pan. The red arrows show the microbial mats and the black arrow shows the sediment. (**B**–**D**) demonstrate the variation between the water content of the samples and are representative of samples MMA, MMB and MMC, respectively.

**Figure 3 microorganisms-12-00147-f003:**
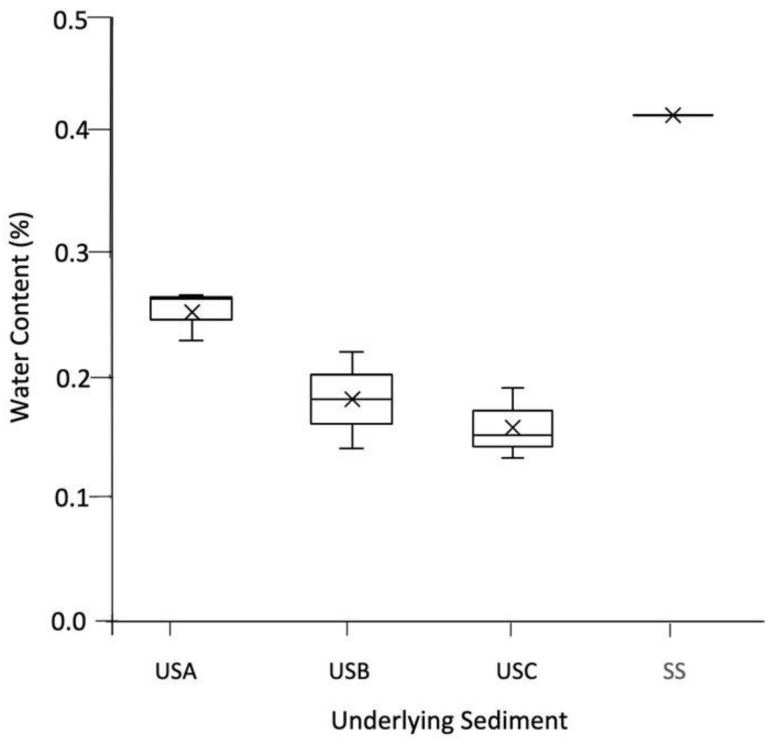
The water content (%) of the underlying sediment. The values reported are the means of three independent measurements, and the standard error associated with these determinations is shown.

**Figure 4 microorganisms-12-00147-f004:**
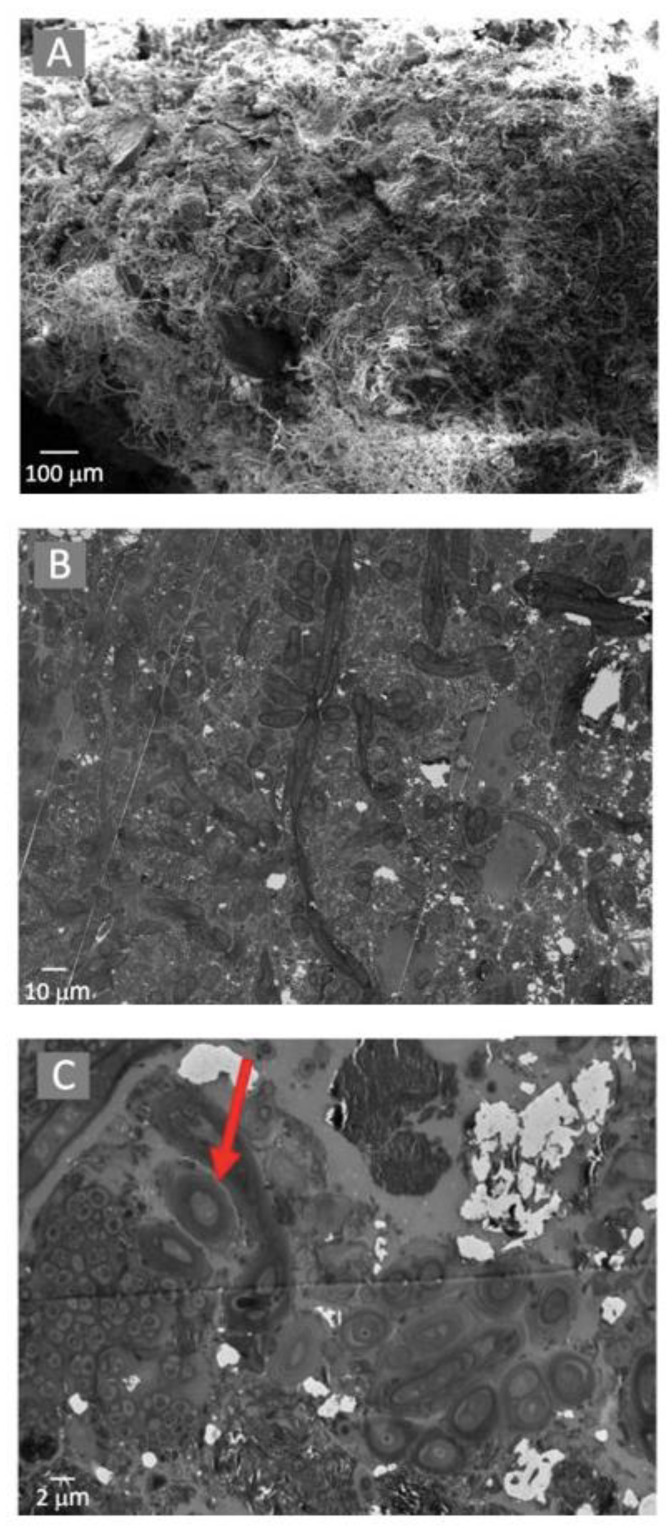
Electron microscopy observations of the microbial mats using a SEM with EDS (**A**) and TEM (**B**). Potential Extracellular Polysaccharide Substrate was observed using TEM (demonstrated by the arrow in (**C**)).

**Figure 5 microorganisms-12-00147-f005:**
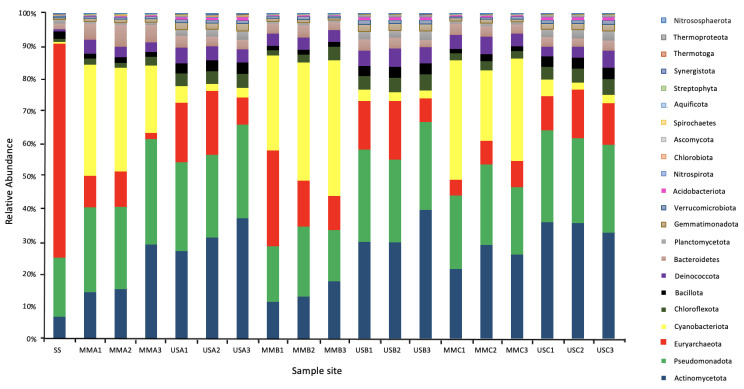
Microbial community composition at the Phylum level of the Microbial Mats (MM), Underlying Sediments (US) and the surface sediment (SS). The Phylum were assigned based on conserved genes in the metagenomic sequencing data set.

**Figure 6 microorganisms-12-00147-f006:**
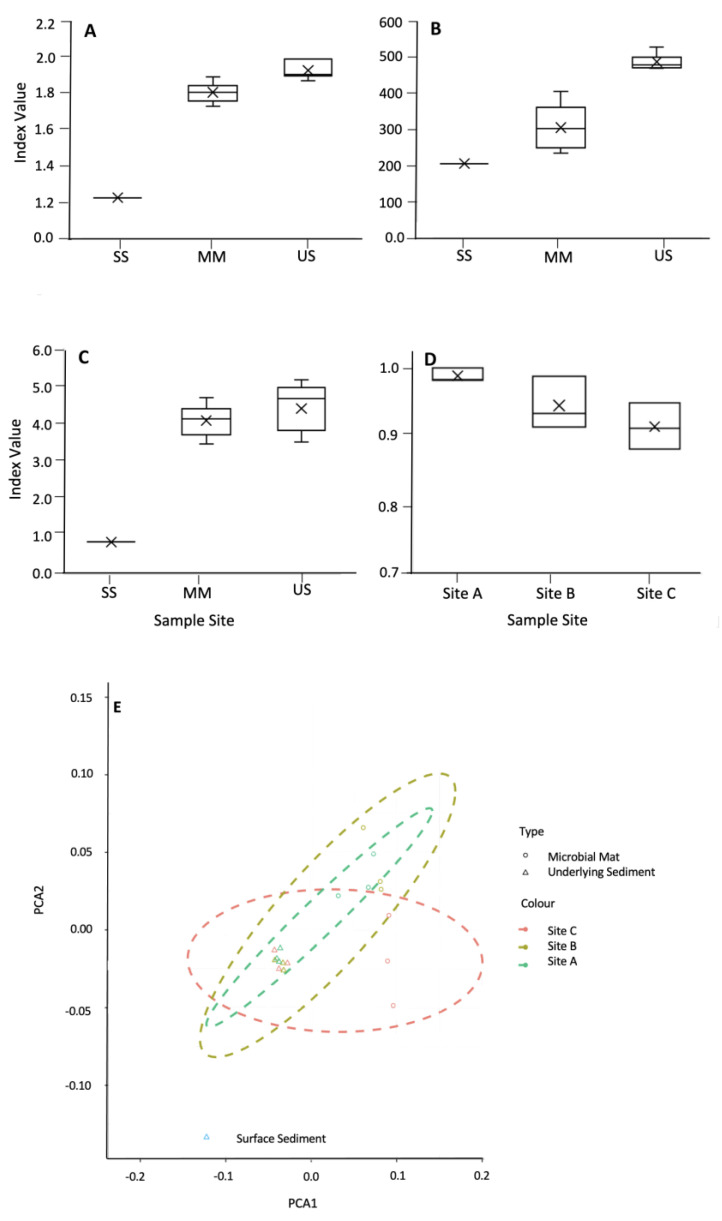
Shannon index (**A**), α-diversity (**B**), and Simpson index (**C**) for the microbial mats, their underlying sediment, and the surface sediment. (**D**) Sørensen index for the similarity between the mats and their underlying sediment. (**E**) Ordination plot using NMDA showing the distribution of the microbial mats (open circles) and the underlying sediment (open triangles).

**Table 1 microorganisms-12-00147-t001:** Geochemical characterisation of the microbial mats and sediments from the Ntwetwe Pan.

Sample Name	Water Content (g/g)	pH	TOC (%)	TN (%)	C/N
Microbial Mats					
MMA1	N/A	9.80	2.19	0.29	7.55
MMA2	N/A	9.76	1.06	0.09	11.7
MMA3	N/A	10.1	0.83	0.06	13.8
MMB1	N/A	9.63	0.93	0.08	11.6
MMB2	N/A	9.76	1.29	0.13	9.92
MMB3	N/A	9.73	1.27	0.14	9.07
MMC1	N/A	9.92	0.84	0.06	14.01
MMC2	N/A	10.01	N/A	N/A	N/A
MMC3	N/A	10.15	3.45	0.37	9.32
Underlying Sediment					
USA1	0.264	9.87	0.71	0.06	11.8
USA2	0.227	9.97	1.35	0.12	11.2
USA3	0.261	9.51	0.45	0.02	22.5
USB1	0.218	9.31	0.74	0.06	12.3
USB2	0.139	9.43	0.86	0.05	17.2
USB3	0.179	9.72	1.50	0.13	11.5
USC1	0.151	9.98	0.74	0.04	18.5
USC2	0.132	9.90	BDL	BDL	N/A
USC3	0.188	10.0	0.77	0.06	12.83
Control					
Surface sediment	0.4103	10.28	0.18	BDL	N/A

BDL—Below the Detection Limits (0.1 for TOC, 0.01 for N). N/A: Not applicable for water content in thin biofilm microbial mats.

## Data Availability

Raw data were submitted to NCBI under accession numbers SRR27256811-SRR27256830.

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
