# Peer review of "Diversity of Microbial Mats in the Makgadikgadi Salt Pans, Botswana"

_microorganisms, 2024, doi:10.3390/microorganisms12010147_

Round 1

Reviewer 1 Report

Comments and Suggestions for Authors

The manuscript describes microbial mats diversity found in the Makgadikgadi Salt Pans. Overall, I believe the research is sound and very interesting for the field. However, I believe the methods section must be improved.

Specifically, the line 151, the authors said that they use FragGenScan for protein-coding regions prediction, but there is no description on how taxonomical assignments were done. It is not information regarding the database for comparison, and cutt off. 

Also, in line 189, the authors described a cut off 70% identity to ribosomal sequences, but it is no clear if the sequencing was of ribosomal genes (16S rRNA) or complete unbiassed sequencing.

I think this issues must be corrected for further acceptance for publication

Author Response

Thank you for the time you spent reviewing this paper. We have addressed your comments as follows:

1) We have removed the reference to the FragGenScan and replaced it with the methodology for the taxonomical assignments:

MEGAHIT v1.2.9 performed de novo assembly with default parameters to obtain contigs (min 3000 bp) (Dinghua et al., 2015). Potential rRNA gene sequences were identified by BLAST similarity searches against the M5rna database, which integrates SILVA (Pruesse et al. 2007), Greengenes (DeSantis et al. 2006), and Ribosomal Database Project (RDP) (Cole et al. 2003).

2) We have made it clear in the text that it was ribosomal genes: The overall taxonomic analysis of the metagenomic reads demonstrated a total of 67 Phyla, with a cut-off of 70% identity to a database of ribosomal sequences.

Reviewer 2 Report

Comments and Suggestions for Authors

The authors have presented a rather interesting paper dedicated to the research issues of the African salt marsh consortium, which is interdisciplinary and combines geological, soil science as well as biological knowledge obtained by collecting samples in real conditions. The work is "alive" and interesting to study. In the course of reading, I had a few questions or comments that I want to ask here.

Lines 85-87: please specify how long, more precisely, not longer than how long the samples were stored at +4 and at -20C.

Line 145: I couldn't find the data, although I don't doubt that the authors deposited it, maybe it makes sense to provide a link here for easier access to it? In general, there have been no such problems with NCBI.

Line 169: "(AS)A" looks weird to me.

Line 187: Judging from the length of 149 +- 8 bp reads were trimmed? Also most likely the reads were paired? In any case, I would like to see this information in the corresponding part of the "methods" section.

Line 196: Correct "Pseudomonodata" to Pseudomonadota

Line 198: What exactly should the ratio that so attracted the authors, Actinobacteriota / PseudomonADOTa (correct here too!), tell us? Why these particular strains? What does it show? Why is there no mention of this ratio in the discussion?

Line 207: The control sample is mentioned for the first time. Although I believe that how was it obtained? under what conditions? why do the authors call it a control? what is the difference with normal samples? -- all of this should be in the "methods" section. Can it be called a control if it did not contain a microbial mat?

Lines 211-219: the authors round to two decimal places, but it is probably more important to give 3-digit values: 0.0246, 3.17, 29.2, etc.

Line 274: ChlorOflexota, BacterOidota... CHECK EVERYTHING, YOU HEAR ME, EVERYTHING!!! NAMES OF TAXONS, geographical objects. And don't forget the names of the authors!

lines 305-307: here too there is information about the control, but no information about how it was taken. I have a suspicion that this is not a control at all. What distinguishes it from the irrelevant sample? Why is it a control?

In general, the careless attitude towards biological names is irritating!

Author Response

Please see the attached review.
